# Greedy Subspace Clustering

**Dohyung Park**
Dept. of Electrical and Computer Engineering
The University of Texas at Austin
dhpark@utexas.edu

**Constantine Caramanis**
Dept. of Electrical and Computer Engineering
The University of Texas at Austin
constantine@utexas.edu

**Sujay Sanghavi**
Dept. of Electrical and Computer Engineering
The University of Texas at Austin
sanghavi@mail.utexas.edu

## Abstract

We consider the problem of subspace clustering: given points that lie on or near the *union* of many low-dimensional linear subspaces, recover the subspaces. To this end, one first identifies sets of points close to the same subspace and uses the sets to estimate the subspaces. As the geometric structure of the clusters (linear subspaces) forbids proper performance of general distance based approaches such as $K$-means, many model-specific methods have been proposed. In this paper, we provide new simple and efficient algorithms for this problem. Our statistical analysis shows that the algorithms are guaranteed exact (perfect) clustering performance under certain conditions on the number of points and the affinity between subspaces. These conditions are weaker than those considered in the standard statistical literature. Experimental results on synthetic data generated from the standard unions of subspaces model demonstrate our theory. We also show that our algorithm performs competitively against state-of-the-art algorithms on real-world applications such as motion segmentation and face clustering, with much simpler implementation and lower computational cost.

## 1 Introduction

Subspace clustering is a classic problem where one is given points in a high-dimensional ambient space and would like to approximate them by a *union* of lower-dimensional linear subspaces. In particular, each subspace contains a subset of the points. This problem is hard because one needs to jointly find the subspaces, and the points corresponding to each; the data we are given are unlabeled. The unions of subspaces model naturally arises in settings where data from multiple latent phenomena are mixed together and need to be separated. Applications of subspace clustering include motion segmentation [23], face clustering [8], gene expression analysis [10], and system identification [22]. In these applications, data points with the same label (e.g., face images of a person under varying illumination conditions, feature points of a moving rigid object in a video sequence) lie on a low-dimensional subspace, and the mixed dataset can be modeled by unions of subspaces. For detailed description of the applications, we refer the readers to the reviews [10, 20] and references therein.

There is now a sizable literature on empirical methods for this particular problem and some statistical analysis as well. Many recently proposed methods, which perform remarkably well and have theoretical guarantees on their performances, can be characterized as involving two steps: *(a)* finding a "neighborhood" for each data point, and *(b)* finding the subspaces and/or clustering the points given these neighborhoods. Here, neighbors of a point are other points that the algorithm estimates to lie on the same subspace as the point (and not necessarily just closest in Euclidean distance).

| Algorithm | What is guaranteed | Subspace condition | Conditions for: | |
|---|---|---|---|---|
| | | | Fully random model | Semi-random model |
| SSC [4, 16] | Correct neighborhoods | None | $\frac{d}{p} = O(\frac{\log(n/d)}{\log(nL)})$ | $\max \text{aff} = O(\frac{\sqrt{\log(n/d)}}{\log(nL)})$ |
| LRR [14] | Exact clustering | No intersection | - | - |
| SSC-OMP [3] | Correct neighborhoods | No intersection | - | - |
| TSC [6, 7] | Exact clustering | None | $\frac{d}{p} = O(\frac{1}{\log(nL)})$ | $\max \text{aff} = O(\frac{1}{\log(nL)})$ |
| LRSSC [24] | Correct neighborhoods | None | $\frac{d}{p} = O(\frac{1}{\log(nL)})$ | - |
| NSN+GSR | Exact clustering | None | $\frac{d}{p} = O(\frac{\log n}{\log(ndL)})$ | $\max \text{aff} = O(\sqrt{\frac{\log n}{(\log dL) \cdot \log(ndL)}})$ |
| NSN+Spectral | Exact clustering | None | $\frac{d}{p} = O(\frac{\log n}{\log(ndL)})$ | - |

Table 1: Subspace clustering algorithms with theoretical guarantees. LRR and SSC-OMP have only deterministic guarantees, not statistical ones. In the two standard statistical models, there are $n$ data points on each of $L$ $d$-dimensional subspaces in $\mathbb{R}^p$. For the definition of $\max \text{aff}$, we refer the readers to Section 3.1.

**Our contributions:** In this paper we devise **new algorithms** for each of the two steps above; *(a)* we develop a new method, Nearest Subspace Neighbor (NSN), to determine a neighborhood set for each point, and *(b)* a new method, Greedy Subspace Recovery (GSR), to recover subspaces from given neighborhoods. Each of these two methods can be used in conjunction with other methods for the corresponding other step; however, in this paper we focus on two algorithms that use NSN followed by GSR and Spectral clustering, respectively. Our main result is establishing **statistical guarantees for exact clustering with general subspace conditions**, in the standard models considered in recent analytical literature on subspace clustering. Our condition for exact recovery is weaker than the conditions of other existing algorithms that only guarantee *correct neighborhoods*[1], which do not always lead to correct clustering. We provide numerical results which demonstrate our theory. We also show that for the real-world applications our algorithm performs competitively against those of state-of-the-art algorithms, but the computational cost is much lower than them. Moreover, our algorithms are much simpler to implement.

## 1.1 Related work

The problem was first formulated in the data mining community [10]. Most of the related work in this field assumes that an underlying subspace is parallel to some canonical axes. Subspace clustering for unions of arbitrary subspaces is considered mostly in the machine learning and the computer vision communities [20]. Most of the results from those communities are based on empirical justification. They provided algorithms derived from theoretical intuition and showed that they perform empirically well with practical dataset. To name a few, GPCA [21], Spectral curvature clustering (SCC) [2], and many iterative methods [1, 19, 26] show their good empirical performance for subspace clustering. However, they lack theoretical analysis that guarantees exact clustering.

As described above, several algorithms with a common structure are recently proposed with both theoretical guarantees and remarkable empirical performance. Elhamifar and Vidal [4] proposed an algorithm called Sparse Subspace Clustering (SSC), which uses $\ell_1$-minimization for neighborhood construction. They proved that if the subspaces have *no intersection*[2], SSC always finds a correct neighborhood matrix. Later, Soltanolkotabi and Candes [16] provided a statistical guarantee of the algorithm for subspaces with intersection. Dyer et al. [3] proposed another algorithm called SSC-OMP, which uses Orthogonal Matching Pursuit (OMP) instead of $\ell_1$-minimization in SSC. Another algorithm called Low-Rank Representation (LRR) which uses nuclear norm minimization is proposed by Liu et al. [14]. Wang et al. [24] proposed an hybrid algorithm, Low-Rank and Sparse Subspace Clustering (LRSSC), which involves both $\ell_1$-norm and nuclear norm. Heckel and Bölcskei [6] presented Thresholding based Subspace Clustering (TSC), which constructs neighborhoods based on the inner products between data points. All of these algorithms use spectral clustering for the clustering step.

The analysis in those papers focuses on neither exact recovery of the subspaces nor exact clustering in general subspace conditions. SSC, SSC-OMP, and LRSSC only guarantee correct neighborhoods which do not always lead to exact clustering. LRR guarantees exact clustering only when

the subspaces have no intersections. In this paper, we provide novel algorithms that guarantee exact clustering in general subspace conditions. When we were preparing this manuscript, it is proved that TSC guarantees exact clustering under certain conditions [7], but the conditions are stricter than ours. (See Table 1)

## 1.2 Notation

There is a set of $N$ data points in $\mathbb{R}^p$, denoted by $\mathcal{Y} = \{y_1, \ldots, y_N\}$. The data points are lying on or near a union of $L$ subspaces $\mathcal{D} = \cup_{i=1}^{L} \mathcal{D}_i$. Each subspace $\mathcal{D}_i$ is of dimension $d_i$ which is smaller than $p$. For each point $y_j$, $w_j$ denotes the index of the nearest subspace. Let $N_i$ denote the number of points whose nearest subspace is $\mathcal{D}_i$, i.e., $N_i = \sum_{j=1}^{N} \mathbb{I}_{w_j=i}$. Throughout this paper, sets and subspaces are denoted by calligraphic letters. Matrices and key parameters are denoted by letters in upper case, and vectors and scalars are denoted by letters in lower case. We frequently denote the set of $n$ indices by $[n] = \{1, 2, \ldots, n\}$. As usual, $\mathrm{span}\{\cdot\}$ denotes a subspace spanned by a set of vectors. For example, $\mathrm{span}\{v_1, \ldots, v_n\} = \{v : v = \sum_{i=1}^{n} \alpha_i v_i, \alpha_1, \ldots, \alpha_n \in \mathbb{R}\}$. $\mathrm{Proj}_{\mathcal{U}} y$ is defined as the projection of $y$ onto subspace $\mathcal{U}$. That is, $\mathrm{Proj}_{\mathcal{U}} y = \arg\min_{u \in \mathcal{U}} \|y - u\|_2$. $\mathbb{I}\{\cdot\}$ denotes the indicator function which is one if the statement is true and zero otherwise. Finally, $\bigoplus$ denotes the direct sum.

## 2 Algorithms

We propose two algorithms for subspace clustering as follows.

- NSN+GSR : Run Nearest Subspace Neighbor (NSN) to construct a neighborhood matrix $W \in \{0,1\}^{N \times N}$, and then run Greedy Subspace Recovery (GSR) for $W$.
- NSN+Spectral : Run Nearest Subspace Neighbor (NSN) to construct a neighborhood matrix $W \in \{0,1\}^{N \times N}$, and then run spectral clustering for $Z = W + W^\top$.

### 2.1 Nearest Subspace Neighbor (NSN)

NSN approaches the problem of *finding neighbor points most likely to be on the same subspace* in a greedy fashion. At first, given a point $y$ without any other knowledge, the one single point that is most likely to be a neighbor of $y$ is the nearest point of the line $\mathrm{span}\{y\}$. In the following steps, if we have found a few correct neighbor points (lying on the same true subspace) and have no other knowledge about the true subspace and the rest of the points, then the most potentially correct point is the one closest to the subspace spanned by the correct neighbors we have. This motivates us to propose NSN described in the following.

---

**Algorithm 1** Nearest Subspace Neighbor (NSN)

---

**Input:** A set of $N$ samples $\mathcal{Y} = \{y_1, \ldots, y_N\}$, The number of required neighbors $K$, Maximum subspace dimension $k_{\max}$.
**Output:** A neighborhood matrix $W \in \{0,1\}^{N \times N}$
    $y_i \leftarrow y_i / \|y_i\|_2, \forall i \in [N]$                                  ▷ Normalize magnitudes
    **for** $i = 1, \ldots, N$ **do**                             ▷ Run NSN for each data point
        $\mathcal{I}_i \leftarrow \{i\}$
        **for** $k = 1, \ldots, K$ **do**         ▷ Iteratively add the closest point to the current subspace
            **if** $k \leq k_{\max}$ **then**
                $\mathcal{U} \leftarrow \mathrm{span}\{y_j : j \in \mathcal{I}_i\}$
            **end if**
            $j^* \leftarrow \arg\max_{j \in [N] \setminus \mathcal{I}_i} \|\mathrm{Proj}_{\mathcal{U}} y_j\|_2$
            $\mathcal{I}_i \leftarrow \mathcal{I}_i \cup \{j^*\}$
        **end for**
        $W_{ij} \leftarrow \mathbb{I}_{j \in \mathcal{I}_i \text{ or } y_j \in \mathcal{U}}, \ \forall j \in [N]$       ▷ Construct the neighborhood matrix
    **end for**

---

NSN collects $K$ neighbors sequentially for each point. At each step $k$, a $k$-dimensional subspace $\mathcal{U}$ spanned by the point and its $k-1$ neighbors is constructed, and the point closest to the subspace is

newly collected. After $k \geq k_{\max}$, the subspace $\mathcal{U}$ constructed at the $k_{\max}$th step is used for collecting neighbors. At last, if there are more points lying on $\mathcal{U}$, they are also counted as neighbors. The subspace $\mathcal{U}$ can be stored in the form of a matrix $U \in \mathbb{R}^{p \times \dim(\mathcal{U})}$ whose columns form an orthonormal basis of $\mathcal{U}$. Then $\|\text{Proj}_{\mathcal{U}} y_j\|_2$ can be computed easily because it is equal to $\|U^\top y_j\|_2$. While a naive implementation requires $O(K^2 p N^2)$ computational cost, this can be reduced to $O(K p N^2)$, and the faster implementation is described in Section A.1. We note that this computational cost is much lower than that of the convex optimization based methods (e.g., SSC [4] and LRR [14]) which solve a convex program with $N^2$ variables and $pN$ constraints.

NSN for subspace clustering shares the same philosophy with Orthogonal Matching Pursuit (OMP) for sparse recovery in the sense that it incrementally picks the point (dictionary element) that is the most likely to be correct, assuming that the algorithms have found the correct ones. In subspace clustering, that point is the one closest to the subspace spanned by the currently selected points, while in sparse recovery it is the one closest to the residual of linear regression by the selected points. In the sparse recovery literature, the performance of OMP is shown to be comparable to that of Basis Pursuit ($\ell_1$-minimization) both theoretically and empirically [18, 11]. One of the contributions of this work is to show that this high-level intuition is indeed born out, provable, as we show that NSN also performs well in collecting neighbors lying on the same subspace.

## 2.2 Greedy Subspace Recovery (GSR)

Suppose that NSN has found correct neighbors for a data point. How can we check if they are indeed correct, that is, lying on the same true subspace? One natural way is to count the number of points close to the subspace spanned by the neighbors. If they span one of the true subspaces, then many other points will be lying on the span. If they do not span any true subspaces, few points will be close to it. This fact motivates us to use a greedy algorithm to recover the subspaces. Using the neighborhood constructed by NSN (or some other algorithm), we recover the $L$ subspaces. If there is a neighborhood set containing only the points on the same subspace for each subspace, the algorithm successfully recovers the unions of the true subspaces exactly.

---

**Algorithm 2** Greedy Subspace Recovery (GSR)

---

**Input:** $N$ points $\mathcal{Y} = \{y_1, \ldots, y_N\}$, A neighborhood matrix $W \in \{0, 1\}^{N \times N}$, Error bound $\epsilon$
**Output:** Estimated subspaces $\hat{\mathcal{D}} = \cup_{l=1}^{L} \hat{D}_l$. Estimated labels $\hat{w}_1, \ldots, \hat{w}_N$
$\quad y_i \leftarrow y_i / \|y_i\|_2, \forall i \in [N]$ $\hfill \triangleright$ Normalize magnitudes
$\quad \mathcal{W}_i \leftarrow \text{Top-}d\{y_j : W_{ij} = 1\}, \forall i \in [N]$ $\triangleright$ Estimate a subspace using the neighbors for each point
$\quad \mathcal{I} \leftarrow [N]$
$\quad \textbf{while } \mathcal{I} \neq \emptyset \textbf{ do}$ $\hfill \triangleright$ Iteratively pick the best subspace estimates
$\quad\quad i^* \leftarrow \arg\max_{i \in \mathcal{I}} \sum_{j=1}^{N} \mathbb{I}\{\|\text{Proj}_{\mathcal{W}_i} y_j\|_2 \geq 1 - \epsilon\}$
$\quad\quad \hat{\mathcal{D}}_l \leftarrow \hat{\mathcal{W}}_{i^*}$
$\quad\quad \mathcal{I} \leftarrow \mathcal{I} \setminus \{j : \|\text{Proj}_{\mathcal{W}_{i^*}} y_j\|_2 \geq 1 - \epsilon\}$
$\quad \textbf{end while}$
$\quad \hat{w}_i \leftarrow \arg\max_{l \in [L]} \|\text{Proj}_{\hat{D}_l} y_i\|_2, \forall i \in [N]$ $\hfill \triangleright$ Label the points using the subspace estimates

---

Recall that the matrix $W$ contains the labelings of the points, so that $W_{ij} = 1$ if point $i$ is assigned to subspace $j$. Top-$d\{y_j : W_{ij} = 1\}$ denotes the $d$-dimensional principal subspace of the set of vectors $\{y_j : W_{ij} = 1\}$. This can be obtained by taking the first $d$ left singular vectors of the matrix whose columns are the vector in the set. If there are only $d$ vectors in the set, Gram-Schmidt orthogonalization will give us the subspace. As in NSN, it is efficient to store a subspace $\mathcal{W}_i$ in the form of its orthogonal basis because we can easily compute the norm of a projection onto the subspace.

Testing a candidate subspace by counting the number of near points has already been considered in the subspace clustering literature. In [25], the authors proposed to run RANdom SAmple Consensus (RANSAC) iteratively. RANSAC randomly selects a few points and checks if there are many other points near the subspace spanned by the collected points. Instead of randomly choosing sample points, GSR receives some candidate subspaces (in the form of sets of points) from NSN (or possibly some other algorithm) and selects subspaces in a greedy way as specified in the algorithm above.

# 3 Theoretical results

We analyze our algorithms in two standard noiseless models. The main theorems present sufficient conditions under which the algorithms cluster the points exactly with high probability. For simplicity of analysis, we assume that every subspace is of the same dimension, and the number of data points on each subspace is the same, i.e., $d \triangleq d_1 = \cdots = d_L$, $n \triangleq N_1 = \cdots = N_L$. We assume that $d$ is known to the algorithm. Nonetheless, our analysis can extend to the general case.

## 3.1 Statistical models

We consider two models which have been used in the subspace clustering literature:

- Fully random model: The subspaces are drawn iid uniformly at random, and the points are also iid randomly generated.

- Semi-random model: The subspaces are arbitrarily determined, but the points are iid randomly generated.

Let $D_i \in \mathbb{R}^{p \times d}, i \in [L]$ be a matrix whose columns form an orthonormal basis of $\mathcal{D}_i$. An important measure that we use in the analysis is the *affinity* between two subspaces, defined as

$$\mathrm{aff}(i,j) \triangleq \frac{\|D_i^\top D_j\|_F}{\sqrt{d}} = \sqrt{\frac{\sum_{k=1}^d \cos^2 \theta_k^{i,j}}{d}} \in [0,1],$$

where $\theta_k^{i,j}$ is the $k$th principal angle between $\mathcal{D}_i$ and $\mathcal{D}_j$. Two subspaces $\mathcal{D}_i$ and $\mathcal{D}_j$ are identical if and only if $\mathrm{aff}(i,j) = 1$. If $\mathrm{aff}(i,j) = 0$, every vector on $\mathcal{D}_i$ is orthogonal to any vectors on $\mathcal{D}_j$. We also define the maximum affinity as

$$\max \mathrm{aff} \triangleq \max_{i,j \in [L], i \neq j} \mathrm{aff}(i,j) \in [0,1].$$

There are $N = nL$ points, and there are $n$ points exactly lying on each subspace. We assume that each data point $y_i$ is drawn iid uniformly at random from $\mathbb{S}^{p-1} \cap \mathcal{D}_{w_i}$ where $\mathbb{S}^{p-1}$ is the unit sphere in $\mathbb{R}^p$. Equivalently,

$$y_i = D_{w_i} x_i, \quad x_i \sim \mathrm{Unif}(\mathbb{S}^{d-1}), \quad \forall i \in [N].$$

As the points are generated randomly on their corresponding subspaces, there are no points lying on an intersection of two subspaces, almost surely. This implies that *with probability one the points are clustered correctly provided that the true subspaces are recovered exactly.*

## 3.2 Main theorems

The first theorem gives a statistical guarantee for the fully random model.

**Theorem 1** *Suppose $L$ $d$-dimensional subspaces and $n$ points on each subspace are generated in the fully random model with $n$ polynomial in $d$. There are constants $C_1, C_2 > 0$ such that if*

$$\frac{n}{d} > C_1 \left( \log \frac{ne}{d\delta} \right)^2, \quad \frac{d}{p} < \frac{C_2 \log n}{\log(ndL\delta^{-1})}, \tag{1}$$

*then with probability at least $1 - \frac{3L\delta}{1-\delta}$, NSN+GSR[3] clusters the points exactly. Also, there are other constants $C_1', C_2' > 0$ such that if (1) with $C_1$ and $C_2$ replaced by $C_1'$ and $C_2'$ holds then NSN+Spectral[4] clusters the points exactly with probability at least $1 - \frac{3L\delta}{1-\delta}$. $e$ is the exponential constant.*

Our sufficient conditions for exact clustering explain when subspace clustering becomes easy or difficult, and they are consistent with our intuition. For NSN to find correct neighbors, the points on the same subspace should be many enough so that they look like lying on a subspace. This condition is spelled out in the first inequality of (1). We note that the condition holds even when $n/d$ is a constant, i.e., $n$ is linear in $d$. The second inequality implies that the dimension of the subspaces should not be too high for subspaces to be distinguishable. If $d$ is high, the random subspaces are more likely to be close to each other, and hence they become more difficult to be distinguished. However, as $n$ increases, the points become dense on the subspaces, and hence it becomes easier to identify different subspaces.

Let us compare our result with the conditions required for success in the fully random model in the existing literature. In [16], it is required for SSC to have correct neighborhoods that $n$ should be superlinear in $d$ when $d/p$ fixed. In [6, 24], the conditions on $d/p$ becomes worse as we have more points. On the other hand, our algorithms are guaranteed exact clustering of the points, and the sufficient condition is order-wise at least as good as the conditions for correct neighborhoods by the existing algorithms (See Table 1). Moreover, exact clustering is guaranteed even when $n$ is linear in $d$, and $d/p$ fixed.

For the semi-random model, we have the following general theorem.

**Theorem 2** *Suppose $L$ $d$-dimensional subspaces are arbitrarily chosen, and $n$ points on each subspace are generated in the semi-random model with $n$ polynomial in $d$. There are constants $C_1, C_2 > 0$ such that if*

$$\frac{n}{d} > C_1 \left( \log \frac{ne}{d\delta} \right)^2, \quad \max \text{aff} < \sqrt{\frac{C_2 \log n}{\log(dL\delta^{-1}) \cdot \log(ndL\delta^{-1})}}. \qquad (2)$$

*then with probability at least $1 - \frac{3L\delta}{1-\delta}$, NSN+GSR[5] clusters the points exactly.*

In the semi-random model, the sufficient condition does not depend on the ambient dimension $p$. When the affinities between subspaces are fixed, and the points are exactly lying on the subspaces, the difficulty of the problem does not depend on the ambient dimension. It rather depends on $\max \text{aff}$, which measures how close the subspaces are. As they become closer to each other, it becomes more difficult to distinguish the subspaces. The second inequality of (2) explains this intuition. The inequality also shows that if we have more data points, the problem becomes easier to identify different subspaces.

Compared with other algorithms, NSN+GSR is guaranteed exact clustering, and more importantly, the condition on $\max \text{aff}$ improves as $n$ grows. This remark is consistent with the practical performance of the algorithm which improves as the number of data points increases, while the existing guarantees of other algorithms are not. In [16], correct neighborhoods in SSC are guaranteed if $\max \text{aff} = O(\sqrt{\log(n/d)}/\log(nL))$. In [6], exact clustering of TSC is guaranteed if $\max \text{aff} = O(1/\log(nL))$. However, these algorithms perform empirically better as the number of data points increases.

## 4    Experimental results

In this section, we empirically compare our algorithms with the existing algorithms in terms of clustering performance and computational time (on a single desktop). For NSN, we used the fast implementation described in Section A.1. The compared algorithms are $K$-means, $K$-flats[6], SSC, LRR, SCC, TSC[7], and SSC-OMP[8]. The numbers of replicates in $K$-means, $K$-flats, and the $K$-

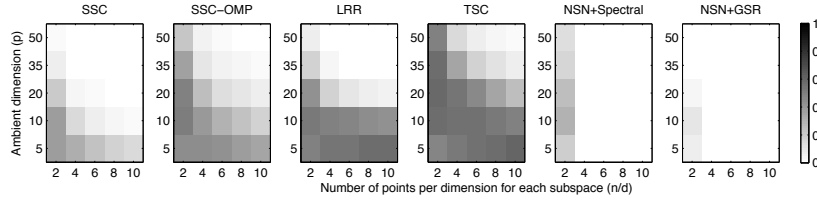

Figure 1: CE of algorithms on 5 random $d$-dimensional subspaces and $n$ random points on each subspace. The figures shows CE for different numbers of $n/d$ and ambient dimension $p$. $d/p$ is fixed to be $3/5$. Brighter cells represent that less data points are clustered incorrectly.

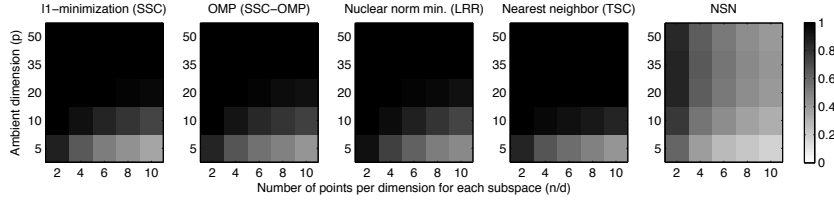

Figure 2: NSE for the same model parameters as those in Figure 1. Brighter cells represent that more data points have all correct neighbors.

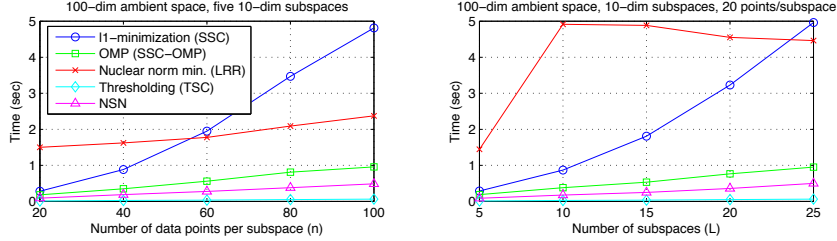

Figure 3: Average computational time of the neighborhood selection algorithms

means used in the spectral clustering are all fixed to 10. The algorithms are compared in terms of *Clustering error (CE)* and *Neighborhood selection error (NSE)*, defined as

$$\text{(CE)} = \min_{\pi \in \Pi_L} \frac{1}{N} \sum_{i=1}^{N} \mathbb{I}(w_i \neq \pi(\hat{w}_i)), \quad \text{(NSE)} = \frac{1}{N} \sum_{i=1}^{N} \mathbb{I}(\exists j : W_{ij} \neq 0, w_i \neq w_j)$$

where $\Pi_L$ is the permutation space of $[L]$. CE is the proportion of incorrectly labeled data points. Since clustering is invariant up to permutation of label indices, the error is equal to the minimum disagreement over the permutation of label indices. NSE measures the proportion of the points which do not have all correct neighbors.[9]

## 4.1 Synthetic data

We compare the performances on synthetic data generated from the fully random model. In $\mathbb{R}^p$, five $d$-dimensional subspaces are generated uniformly at random. Then for each subspace $n$ unit-norm points are generated iid uniformly at random on the subspace. To see the agreement with the theoretical result, we ran the algorithms under fixed $d/p$ and varied $n$ and $d$. We set $d/p = 3/5$ so that each pair of subspaces has intersection. Figures 1 and 2 show CE and NSE, respectively. Each error value is averaged over 100 trials. Figure 1 indicates that our algorithm clusters the data points better than the other algorithms. As predicted in the theorems, the clustering performance improves

| $L$ | Algorithms | $K$-means | $K$-flats | SSC | LRR | SCC | SSC-OMP(8) | TSC(10) | NSN+Spectral(5) |
|---|---|---|---|---|---|---|---|---|---|
| | Mean CE (%) | 19.80 | 13.62 | 1.52 | 2.13 | 2.06 | 16.92 | 18.44 | 3.62 |
| 2 | Median CE (%) | 17.92 | 10.65 | 0.00 | 0.00 | 0.00 | 12.77 | 16.92 | 0.00 |
| | Avg. Time (sec) | - | 0.80 | 3.03 | 3.42 | 1.28 | 0.50 | 0.50 | 0.25 |
| | Mean CE (%) | 26.10 | 14.07 | 4.40 | 4.03 | 6.37 | 27.96 | 28.58 | 8.28 |
| 3 | Median CE (%) | 20.48 | 14.18 | 0.56 | 1.43 | 0.21 | 30.98 | 29.67 | 2.76 |
| | Avg. Time (sec) | - | 1.89 | 5.39 | 4.05 | 2.16 | 0.82 | 1.15 | 0.51 |

Table 2: CE and computational time of algorithms on Hopkins155 dataset. $L$ is the number of clusters (motions). The numbers in the parentheses represent the number of neighbors for each point collected in the corresponding algorithms.

| $L$ | Algorithms | $K$-means | $K$-flats | SSC | SSC-OMP | TSC | NSN+Spectral |
|---|---|---|---|---|---|---|---|
| | Mean CE (%) | 45.98 | 37.62 | 1.77 | 4.45 | 11.84 | 1.71 |
| 2 | Median CE (%) | 47.66 | 39.06 | 0.00 | 1.17 | 1.56 | 0.78 |
| | Avg. Time (sec) | - | 15.78 | 37.72 | 0.45 | 0.33 | 0.78 |
| | Mean CE (%) | 62.55 | 45.81 | 5.77 | 6.35 | 20.02 | 3.63 |
| 3 | Median CE (%) | 63.54 | 47.92 | 1.56 | 2.86 | 15.62 | 3.12 |
| | Avg. Time (sec) | - | 27.91 | 49.45 | 0.76 | 0.60 | 3.37 |
| | Mean CE (%) | 73.77 | 55.51 | 4.79 | 8.93 | 11.90 | 5.81 |
| 5 | Median CE (%) | 74.06 | 56.25 | 2.97 | 5.00 | 33.91 | 4.69 |
| | Avg. Time (sec) | - | 52.90 | 74.91 | 1.41 | 1.17 | 5.62 |
| | Mean CE (%) | 82.68 | 62.72 | 9.43 | 15.32 | 39.48 | 9.82 |
| 10 | Median CE (%) | 82.97 | 62.89 | 8.75 | 17.11 | 39.45 | 9.06 |
| | Avg. Time (sec) | - | 134.0 | 157.5 | 5.26 | 3.17 | 14.73 |

Table 3: CE and computational time of algorithms on Extended Yale B dataset. For each number of clusters (faces) $L$, the algorithms ran over 100 random subsets drawn from the overall 38 clusters.

as the number of points increases. However, it also improves as the dimension of subspaces grows in contrast to the theoretical analysis. We believe that this is because our analysis on GSR is not tight. In Figure 2, we can see that more data points obtain correct neighbors as $n$ increases or $d$ decreases, which conforms the theoretical analysis.

We also compare the computational time of the neighborhood selection algorithms for different numbers of subspaces and data points. As shown in Figure 3, the greedy algorithms (OMP, Thresholding, and NSN) are significantly more scalable than the convex optimization based algorithms ($\ell_1$-minimization and nuclear norm minimization).

## 4.2 Real-world data : motion segmentation and face clustering

We compare our algorithm with the existing ones in the applications of motion segmentation and face clustering. For the motion segmentation, we used Hopkins155 dataset [17], which contains 155 video sequences of 2 or 3 motions. For the face clustering, we used Extended Yale B dataset with cropped images from [5, 13]. The dataset contains 64 images for each of 38 individuals in frontal view and different illumination conditions. To compare with the existing algorithms, we used the set of $48 \times 42$ resized raw images provided by the authors of [4]. The parameters of the existing algorithms were set as provided in their source codes.[10] Tables 2 and 3 show CE and average computational time.[11] We can see that NSN+Spectral performs competitively with the methods with the lowest errors, but much faster. Compared to the other greedy neighborhood construction based algorithms, SSC-OMP and TSC, our algorithm performs significantly better.

## Acknowledgments

The authors would like to acknowledge NSF grants 1302435, 0954059, 1017525, 1056028 and DTRA grant HDTRA1-13-1-0024 for supporting this research. This research was also partially supported by the U.S. Department of Transportation through the Data-Supported Transportation Operations and Planning (D-STOP) Tier 1 University Transportation Center.

## Footnotes

[1]By correct neighborhood, we mean that for each point every neighbor point lies on the same subspace.

[2]By no intersection between subspaces, we mean that they share only the null point.

[3]NSN with $K = k_{max} = d$ followed by GSR with arbitrarily small $\epsilon$.

[4]NSN with $K = k_{max} = d$.

[5]NSN with $K = d - 1$ and $k_{max} = \lceil 2 \log d \rceil$ followed by GSR with arbitrarily small $\epsilon$.

[6]$K$-flats is similar to $K$-means. At each iteration, it computes top-$d$ principal subspaces of the points with the same label, and then labels every point based on its distances to those subspaces.

[7]The MATLAB codes for SSC, LRR, SCC, and TSC are obtained from `http://www.cis.jhu.edu/~ehsan/code.htm`, `https://sites.google.com/site/guangcanliu`, and `http://www.math.duke.edu/~glchen/scc.html`, `http://www.nari.ee.ethz.ch/commth/research/downloads/sc.html`, respectively.

[8]For each data point, OMP constructs a neighborhood for each point by regressing the point on the other points up to $10^{-4}$ accuracy.

[9]For the neighborhood matrices from SSC, LRR, and SSC-OMP, the $d$ points with the maximum weights are regarded as neighbors for each point. For TSC, the $d$ nearest neighbors are collected for each point.

[10]As SSC-OMP and TSC do not have proposed number of parameters for motion segmentation, we found the numbers minimizing the mean CE. The numbers are given in the table.

[11]The LRR code provided by the author did not perform properly with the face clustering dataset that we used. We did not run NSN+GSR since the data points are not well distributed in its corresponding subspaces.

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
