[Supplementary Material]

# A Discussion on implementation issues

## A.1 A faster implementation for NSN

At each step of NSN, the algorithm computes the projections of all points onto a subspace and find one with the largest norm. A naive implementation of the algorithm requires $O(pK^2N^2)$ time complexity.

In fact, we can reduce the complexity to $O(pKN^2)$. Instead of finding the maximum norm of the projections, we can find the maximum squared norm of the projections. Let $\mathcal{U}_k$ be the subspace $\mathcal{U}$ at step $k$. For any data point $y$, we have

$$\|\text{Proj}_{\mathcal{U}_k} y\|_2^2 = \|\text{Proj}_{\mathcal{U}_{k-1}} y\|_2^2 + |u_k^\top y|^2$$

where $u_k$ is the new orthogonal axis added from $\mathcal{U}_{k-1}$ to make $\mathcal{U}_k$. That is, $\mathcal{U}_{k-1} \perp u_k$ and $\mathcal{U}_k = \mathcal{U}_{k-1} \bigoplus u_k$. As $\|\text{Proj}_{\mathcal{U}_{k-1}} y\|_2^2$ is already computed in the $(k-1)$'th step, we do not need to compute it again at step $k$. Based on this fact, we have a faster implementation as described in the following. Note that $P_j$ at the $k$th step is equal to $\|\text{Proj}_{\mathcal{U}_k} y_j\|_2^2$ in the original NSN algorithm.

---

**Algorithm 3** Fast Nearest Subspace Neighbor (F-NSN)

---

**Input:** A set of $N$ samples $\mathcal{Y} = \{y_1, \ldots, y_N\}$, The number of required neighbors $K$, Maximum subspace dimension $k_{max}$.
**Output:** A neighborhood matrix $W \in \{0, 1\}^{N \times N}$
  $y_i \leftarrow y_i / \|y_i\|_2, \forall i \in [N]$
  **for** $i = 1, \ldots, N$ **do**
    $\mathcal{I}_i \leftarrow \{i\}, u_1 \leftarrow y_i$
    $P_j \leftarrow 0, \forall j \in [N]$
    **for** $k = 1, \ldots, K$ **do**
      **if** $k \leq k_{max}$ **then**
        $P_j \leftarrow P_j + \|u_k^\top y_j\|^2, \ \forall j \in [N]$
      **end if**
      $j^* \leftarrow \arg\max_{j \in [N], j \notin \mathcal{I}_i} P_j$
      $\mathcal{I}_i \leftarrow \mathcal{I}_i \cup \{j^*\}$
      **if** $k < k_{max}$ **then**
        $u_{k+1} \leftarrow \dfrac{y_{j^*} - \sum_{l=1}^k (u_l^\top y_{j^*}) u_l}{\|y_{j^*} - \sum_{l=1}^k (u_l^\top y_{j^*}) u_l\|_2}$
      **end if**
    **end for**
    $W_{ij} \leftarrow \mathbb{I}_{j \in \mathcal{I}_i \text{ or } P_j = 1}, \ \forall j \in [N]$
  **end for**

---

## A.2 Estimation of the number of clusters

When $L$ is unknown, it can be estimated at the clustering step. For Spectral clustering, a well-known approach to estimate $L$ is to find a knee point in the singular values of the neighborhood matrix. It is the point where the difference between two consecutive singular values are the largest. For GSR, we do not need to estimate the number of clusters a priori. Once the algorithms finishes, the number of the resulting groups will be our estimate of $L$.

## A.3 Parameter setting

The choices of $K$ and $k_{\max}$ depend on the dimension of the subspaces $d$. If data points are lying exactly on the model subspaces, $K = k_{\max} = d$ is enough for GSR to recover a subspace. In practical situations where the points are near the subspaces, it is better to set $K$ to be larger than $d$. $k_{\max}$ can also be larger than $d$ because the $k_{\max} - d$ additional dimensions, which may be induced from the noise, do no intersect with the other subspaces in practice. For Extended Yale B dataset and Hopkins155 dataset, we found that NSN+Spectral performs well if $K$ is set to be around $2d$.

# B Proofs

## B.1 Proof outline

We describe the first few high-level steps in the proofs of our main theorems. Exact clustering of our algorithms depends on whether NSN can find all correct neighbors for the data points so that the following algorithm (GSR or Spectral clustering) can cluster the points exactly. For NSN+GSR, exact clustering is guaranteed when there is a point on each subspace that have all correct neighbors which are at least $d - 1$. For NSN+Spectral, exact clustering is guaranteed when each data point has only the $n - 1$ other points on the same subspace as neighbors. In the following, we explain why these are true.

### Step 1-1: Exact clustering condition for GSR

The two statistical models have a property that for any $d$-dimensional subspace in $\mathbb{R}^p$ other than the true subspaces $\mathcal{D}_1, \ldots, \mathcal{D}_L$ the probability of any points lying on the subspace is zero. Hence, we claim the following.

**Fact 3 (Best $d$-dimensional fit)** *With probability one, the true subspaces $\mathcal{D}_1, \ldots, \mathcal{D}_L$ are the $L$ subspaces containing the most points among the set of all possible $d$-dimensional subspaces.*

Then it suffices for each subspace to have one point whose neighbors are $d - 1$ all correct points on the same subspace. This is because the subspace spanned by those $d$ points is almost surely identical to the true subspace they are lying on, and that subspace will be picked by GSR.

**Fact 4** *If NSN with $K \geq d - 1$ finds all correct neighbors for at least one point on each subspace, GSR recovers all the true subspaces and clusters the data points exactly with probability one.*

In the following steps, we consider one data point for each subspace. We show that NSN with $K = k_{max} = d$ finds all correct neighbors for the point with probability at least $1 - \frac{3\delta}{1-\delta}$. Then the union bound and Fact 4 establish exact clustering with probability at least $1 - \frac{3L\delta}{1-\delta}$.

### Step 1-2: Exact clustering condition for spectral clustering

It is difficult to analyze spectral clustering for the resulting neighborhood matrix of NSN. A trivial case for a neighborhood matrix to result in exact clustering is when the points on the same subspace form a single fully connected component. If NSN with $K = k_{max} = d$ finds all correct neighbors for every data point, the subspace $\mathcal{U}$ at the last step ($k = K$) is almost surely identical to the true subspace that the points lie on. Hence, the resulting neighborhood matrix $W$ form $L$ fully connected components each of which contains all of the points on the same subspace.

In the rest of the proof, we show that if (1) holds, NSN finds all correct neighbors for a fixed point with probability $1 - \frac{3\delta}{1-\delta}$. Let us assume that this is true. If (1) with $C_1$ and $C_2$ replaced by $\frac{C_1}{4}$ and $\frac{C_2}{2}$ holds, we have

$$ n > C_1 d \left( \log \frac{ne}{d(\delta/n)} \right)^2, \quad \frac{d}{p} < \frac{C_2 \log n}{\log(ndL(\delta/n)^{-1})}. $$

Then it follows from the union bound that NSN finds all correct neighbors for all of the $n$ points on each subspace with probability at least $1 - \frac{3L\delta}{1-\delta}$, and hence we obtain $W_{ij} = \mathbb{I}_{w_i = w_j}$ for every $(i, j) \in [N]^2$. Exact clustering is guaranteed.

### Step 2: Success condition for NSN

Now the only proposition that we need to prove is that for each subspace $\mathcal{D}_i$ NSN finds all correct neighbors for a data point (which is a uniformly random unit vector on the subspace) with probability at least $1 - \frac{3\delta}{1-\delta}$. As our analysis is independent of the subspaces, we only consider $\mathcal{D}_1$. Without loss of generality, we assume that $y_1$ lies on $\mathcal{D}_1$ ($w_1 = 1$) and focus on this data point.

When NSN finds neighbors of $y_1$, the algorithm constructs $k_{max}$ subspaces incrementally. At each step $k = 1, \ldots, K$, if the largest projection onto $\mathcal{U}$ of the uncollected points on the same true subspace $\mathcal{D}_1$ is greater than the largest projection among the points on different subspaces, then NSN collects a correct neighbor. In a mathematical expression, we want to satisfy

$$\max_{j:w_j=1,j\notin\mathcal{I}_1} \|\text{Proj}_{\mathcal{U}}y_j\|_2 > \max_{j:w_j\neq1,j\notin\mathcal{I}_1} \|\text{Proj}_{\mathcal{U}}y_j\|_2 \tag{3}$$

for each step of $k = 1, \ldots, K$.

The rest of the proof is to show (1) and (2) lead to (3) with probability $1 - \frac{3\delta}{1-\delta}$ in their corresponding models. It is difficult to prove (3) itself because the subspaces, the data points, and the index set $\mathcal{I}_1$ are all dependent of each other. Instead, we introduce an Oracle algorithm whose success is equivalent to the success of NSN, but the analysis is easier. Then the Oracle algorithm is analyzed using stochastic ordering, bounds on order statistics of random projections, and the measure concentration inequalities for random subspaces. The rest of the proof is provided in Sections B.3 and B.4.

## B.2 Preliminary lemmas

Before we step into the technical parts of the proof, we introduce the main ingredients which will be used. The following lemma is about upper and lower bounds on the order statistics for the projections of iid uniformly random unit vectors.

**Lemma 5** *Let $x_1, \ldots, x_n$ be drawn iid uniformly at random from the $d$-dimensional unit ball $\mathbb{S}^{d-1}$. Let $z_{(n-m+1)}$ denote the $m$'th largest value of $\{z_i \triangleq \|Ax_i\|_2, 1 \leq i \leq n\}$ where $A \in \mathbb{R}^{k\times d}$.*

*a. Suppose that the rows of $A$ are orthonormal to each other. For any $\alpha \in (0,1)$, there exists a constant $C > 0$ such that for $n, m, d, k \in \mathbb{N}$ where*

$$n - m + 1 \geq Cm \left(\log \frac{ne}{m\delta}\right)^2 \tag{4}$$

*we have*

$$z_{(n-m+1)}^2 > \frac{k}{d} + \frac{1}{d} \cdot \min\left\{2\log\left(\frac{n-m+1}{Cm\left(\log\frac{ne}{m\delta}\right)^2}\right), \alpha\sqrt{d-k}\right\} \tag{5}$$

*with probability at least $1 - \delta^m$.*

*b. For any $k \times d$ matrix $A$,*

$$z_{(n-m+1)} < \frac{\|A\|_F}{\sqrt{d}} + \frac{\|A\|_2}{\sqrt{d}} \cdot \left(\sqrt{2\pi} + \sqrt{2\log\frac{ne}{m\delta}}\right) \tag{6}$$

*with probability at least $1 - \delta^m$.*

Lemma 5b can be proved by using the measure concentration inequalities [12]. Not only can they provide inequalities for random unit vectors, they also give us inequalities for random subspaces.

**Lemma 6** *Let the columns of $X \in \mathbb{R}^{d\times k}$ be the orthonormal basis of a $k$-dimensional random subspace drawn uniformly at random in $d$-dimensional space.*

*a. For any matrix $A \in \mathbb{R}^{p\times d}$.*

$$E[\|AX\|_F^2] = \frac{k}{d}\|A\|_F^2$$

*b. [15, 12] If $\|A\|_2$ is bounded, then we have*

$$\Pr\left\{\|AX\|_F > \sqrt{\frac{k}{d}}\|A\|_F + \|A\|_2 \cdot \left(\sqrt{\frac{8\pi}{d-1}} + t\right)\right\} \leq e^{-\frac{(d-1)t^2}{8}}.$$

## B.3 Proof of Theorem 2

Following Section B.1, we show in this section that if (2) holds then NSN finds all correct neighbors for $y_1$ (which is assumed to be on $\mathcal{D}_1$) with probability at least $1 - \frac{3\delta}{1-\delta}$.

### Step 3: NSN Oracle algorithm

Consider the Oracle algorithm in the following. Unlike NSN, this algorithm knows the true label of each data point. It picks the point closest to the current subspace among the points with the same label. Since we assume $w_1 = 1$, the Oracle algorithm for $y_1$ picks a point in $\{y_j : w_j = 1\}$ at every step.

---

**Algorithm 4** NSN Oracle algorithm for $y_1$ (assuming $w_1 = 1$)

---

**Input:** A set of $N$ samples $\mathcal{Y} = \{y_1, \ldots, y_N\}$, The number of required neighbors $K = d - 1$, Maximum subspace dimension $k_{max} = \lceil 2 \log d \rceil$

    $\mathcal{I}_1^{(1)} \leftarrow \{1\}$
    **for** $k = 1, \ldots, K$ **do**
        **if** $k \leq k_{max}$ **then**
            $\mathcal{V}_k \leftarrow \text{span}\{y_j : j \in \mathcal{I}_1^{(k)}\}$
            $j_k^* \leftarrow \arg\max_{j \in [N]: w_j = 1, j \notin \mathcal{I}_1^{(k)}} \|\text{Proj}_{\mathcal{V}_k} y_j\|_2$
        **else**
            $j_k^* \leftarrow \arg\max_{j \in [N]: w_j = 1, j \notin \mathcal{I}_1^{(k)}} \|\text{Proj}_{\mathcal{V}_{k_{max}}} y_j\|_2$
        **end if**
        **if** $\max_{j \in [N]: w_j = 1, j \notin \mathcal{I}_i^{(k)}} \|\text{Proj}_{\mathcal{V}_k} y_j\|_2 \leq \max_{j \in [N]: w_j \neq 1} \|\text{Proj}_{\mathcal{V}_k} y\|_2$ **then**
            Return FAILURE
        **end if**
        $\mathcal{I}_1^{(k+1)} \leftarrow \mathcal{I}_1^{(k)} \cup \{j_k^*\}$
    **end for**
    Return SUCCESS

---

Note that the Oracle algorithm returns failure if and only if the original algorithm picks an incorrect neighbor for $y_1$. The reason is as follows. Suppose that NSN for $y_1$ picks the first incorrect point at step $k$. By the step $k - 1$, correct points have been chosen because they are the nearest points for the subspaces in the corresponding steps. The Oracle algorithm will also pick those points because they are the nearest points among the correct points. Hence $\mathcal{U} \equiv \mathcal{V}_k$. At step $k$, NSN picks an incorrect point as it is the closest to $\mathcal{U}$. The Oracle algorithm will declare failure because that incorrect point is closer than the closest point among the correct points. In the same manner, we see that NSN fails if the Oracle NSN fails. Therefore, we can instead analyze the success of the Oracle algorithm. The success condition is written as

$$\|\text{Proj}_{\mathcal{V}_k} y_{j_k^*}\|_2 > \max_{j \in [N]: w_j \neq 1} \|\text{Proj}_{\mathcal{V}_k} y\|_2, \quad \forall k = 1, \ldots, k_{max},$$

$$\|\text{Proj}_{\mathcal{V}_{k_{max}}} y_{j_k^*}\|_2 > \max_{j \in [N]: w_j \neq 1} \|\text{Proj}_{\mathcal{V}_{k_{max}}} y\|_2, \quad \forall k = k_{max} + 1, \ldots, K. \quad (7)$$

Note that $\mathcal{V}_k$'s are independent of the points $\{y_j : j \in [N], w_j \neq 1\}$. We will use this fact in the following steps.

### Step 4: Lower bounds on the projection of correct points (the LHS of (7))

Let $V_k \in \mathbb{R}^{d \times k}$ be such that the columns of $D_1 V_k$ form an orthogonal basis of $\mathcal{V}_k$. Such a $V_k$ exists because $\mathcal{V}_k$ is a $k$-dimensional subspace of $\mathcal{D}_1$. Then we have

$$\|\text{Proj}_{\mathcal{V}_k} y_{j_k^*}\|_2 = \|V_k^\top D_1^\top D_1 x_{j_k^*}\|_2 = \|V_k^\top x_{j_k^*}\|_2$$

In this step, we obtain lower bounds on $\|V_k^\top x_{j_k^*}\|_2$ for $k \leq k_{max}$ and $\|V_{k_{max}}^\top x_{j_k^*}\|_2$ for $k > k_{max}$.

It is difficult to analyze $\|V_k^\top x_{j_k^*}\|_2$ because $V_k$ and $x_{j_k^*}$ are dependent. We instead analyze another random variable that is stochastically dominated by $\|V_k^\top x_{j_k^*}\|_2^2$. Then we use a high-probability lower bound on that variable which also lower bounds $\|V_k^\top x_{j_k^*}\|_2^2$ with high probability.

Define $P_{k,(m)}$ as the $m$'th largest norm of the projections of $n-1$ iid uniformly random unit vector on $\mathbb{S}^{d-1}$ onto a $k$-dimensional subspace. Since the distribution of the random unit vector is isotropic, the distribution of $P_{k,(m)}$ is identical for any $k$-dimensional subspaces independent of the random unit vectors. We have the following key lemma.

**Lemma 7** $\|V_k^\top x_{j_k^*}\|_2$ *stochastically dominates* $P_{k,(k)}$, *i.e.,*

$$\Pr\{\|V_k^\top x_{j_k^*}\|_2 \geq t\} \geq \Pr\{P_{k,(k)} \geq t\}$$

*for any* $t \geq 0$. *Moreover,* $P_{k,(k)} \geq P_{\hat{k},(k)}$ *for any* $\hat{k} \leq k$.

The proof of the lemma is provided in Appendix B.5. Now we can use the lower bound on $P_{k,(k)}$ given in Lemma 5a to bound $\|V_k^\top x_{j_k^*}\|_2$. Let us pick $\alpha$ and $C$ for which the lemma holds. The first inequality of (2) with $C_1 = C + 1$ leads to $n - d > Cd \left(\log \frac{ne}{d\delta}\right)^2$, and also

$$n - k > Ck \left(\log \frac{ne}{k\delta}\right)^2, \quad \forall k = 1, \ldots, d-1. \tag{8}$$

Hence, it follow from Lemma 5a that for each $k = 1, \ldots, k_{max}$, we have

$$\|V_k^\top x_{j_k^*}\|_2 \geq \frac{k}{d} + \frac{1}{d}\min\left\{2\log\left(\frac{n-k+1}{Ck\left(\log\frac{ne}{k\delta}\right)^2}\right), \alpha\sqrt{d-k}\right\}$$

$$\geq \frac{k}{d} + \frac{1}{d}\min\left\{2\log\left(\frac{n-d}{Cd\left(\log\frac{ne}{\delta}\right)^2}\right), \alpha\sqrt{d-2\log d}\right\} \tag{9}$$

with probability at least $1 - \delta^k$.

For $k > k_{max}$, we want to bound $\|\text{Proj}_{\mathcal{V}_{k_{max}}} y_{j_k^*}\|_2$. We again use Lemma 7 to obtain the bound. Since the condition for the lemma holds as shown in (8), we have

$$\|V_{k_{max}}^\top x_{j_k^*}\|_2 \geq \frac{2\log d}{d} + \frac{1}{d}\min\left\{2\log\left(\frac{n-k+1}{Ck\left(\log\frac{ne}{k\delta}\right)^2}\right), \alpha\sqrt{d-2\log d}\right\}$$

$$\geq \frac{2\log d}{d} + \frac{1}{d}\min\left\{2\log\left(\frac{n-d}{Cd\left(\log\frac{ne}{\delta}\right)^2}\right), \alpha\sqrt{d-2\log d}\right\} \tag{10}$$

with probability at least $1 - \delta^k$, for every $k = k_{max} + 1, \ldots, d-1$.

The union bound gives that (9) and (10) hold for all $k = 1, \ldots, d-1$ simultaneously with probability at least $1 - \frac{\delta}{1-\delta}$.

**Step 5: Upper bounds on the projection of incorrect points (the RHS of (7))**

Since we have $\|\text{Proj}_{\mathcal{V}_k} y_j\|_2 = \|V_k^\top D_1^\top D_{w_j} x_j\|_2$, the RHS of (7) can be written as

$$\max_{j:j\in[N], w_j\neq 1} \|V_k^\top D_1^\top D_{w_j} x_j\|_2 \tag{11}$$

In this step, we want to bound (11) for every $k = 1, \ldots, d-1$ by using the concentration inequalities for $V_k$ and $x_j$. Since $V_k$ and $x_j$ are independent, the inequality for $x_j$ holds for any fixed $V_k$.

It follows from Lemma 5b and the union bound that with probability at least $1 - \delta$,

$$\max_{j:j\in[N],w_j\neq 1}\|V_k^\top D_1^\top D_{w_j}x_j\|_2$$

$$\leq \frac{\max_{l\neq 1}\|V_k^\top D_1^\top D_l\|_F}{\sqrt{d}} + \frac{\max_{l\neq 1}\|V_k^\top D_1^\top D_l\|_2}{\sqrt{d}} \cdot \left(\sqrt{2\pi} + \sqrt{2\log\frac{n(L-1)e}{\delta/d}}\right)$$

$$\leq \frac{\max_{l\neq 1}\|V_k^\top D_1^\top D_l\|_F}{\sqrt{d}} \cdot \left(5 + \sqrt{2\log\frac{ndL}{\delta}}\right)$$

for all $k = 1, \ldots, d-1$. The last inequality follows from the fact $\|V_k^\top D_1^\top D_l\|_2 \leq \|V_k^\top D_1^\top D_l\|_F$. Since $\|V_k^\top D_1^\top D_{w_j}\|_2 \leq \|V_k^\top D_1^\top D_{w_j}\|_F \leq \max_{l\neq 1}\|V_k^\top D_1^\top D_l\|_F$ for every $j$ such that $w_j \neq 1$, we have

$$\max_{j:j\in[N],w_j\neq 1}\|V_k^\top D_1^\top D_{w_j}x_j\|_2 \leq \frac{\max_{l\neq 1}\|V_k^\top D_1^\top D_l\|_F}{\sqrt{d}} \cdot \min\left\{5 + \sqrt{2\log\frac{ndL}{\delta}}, \sqrt{d}\right\}. \quad (12)$$

Now let us consider $\max_{l\neq 1}\|V_k^\top D_1^\top D_l\|_F$. In our statistical model, the new axis added to $\mathcal{V}_k$ at the $k$th step ($u_{k+1}$ in Algorithm 3) is chosen uniformly at random from the subspace in $\mathcal{D}_1$ orthogonal to $\mathcal{V}_k$. Therefore, $V_k$ is a random matrix drawn uniformly from the $d \times k$ Stiefel manifold, and the probability measure is the normalized Haar (rotation-invariant) measure. From Lemma 6b and the union bound, we obtain that with probability at least $1 - \delta/dL$,

$$\|V_k^\top D_1^\top D_l\|_F \leq \sqrt{\frac{k}{d}}\|D_1^\top D_l\|_F + \|D_1^\top D_l\|_2 \cdot \left(\sqrt{\frac{8\pi}{d-1}} + \sqrt{\frac{8}{d-1}\log\frac{dL}{\delta}}\right)$$

$$\leq \|D_1^\top D_l\|_F \cdot \left(\sqrt{\frac{k}{d}} + \sqrt{\frac{8\pi}{d-1}} + \sqrt{\frac{8}{d-1}\log\frac{dL}{\delta}}\right)$$

$$\leq \max\text{aff} \cdot \sqrt{d} \cdot \left(\sqrt{\frac{k}{d}} + \sqrt{\frac{8\pi}{d-1}} + \sqrt{\frac{8}{d-1}\log\frac{dL}{\delta}}\right). \quad (13)$$

The union bound gives that with probability at least $1 - \delta$, $\max_{l\neq 1}\|V_k^\top D_1^\top D_l\|_F$ is also bounded by (13) for every $k = 1, \ldots, k_{max}$.

Putting (13) and (12) together, we obtain

$$\max_{j:j\in[N],w_j\neq 1}\|V_k^\top D_1^\top D_{w_j}x_j\|_2$$

$$\leq \max\text{aff} \cdot \left(\sqrt{\frac{k}{d}} + \sqrt{\frac{8\pi}{d-1}} + \sqrt{\frac{8}{d-1}\log\frac{dL}{\delta}}\right) \cdot \min\left\{5 + \sqrt{2\log\frac{ndL}{\delta}}, \sqrt{d}\right\} \quad (14)$$

for all $k = 1, \ldots, d-1$ with probability at least $1 - 2\delta$.

**Final Step: Proof of the main theorem**

Putting (9), (10), and (14) together, we obtain that if

$$\max\text{aff} < \min_{1\leq k\leq d-1}\frac{\sqrt{\min\{k, 2\log d\} + \min\left\{2\log\left(\frac{n-d}{Cd}\right) - 4\log\log\frac{ne}{\delta}, \alpha\sqrt{d - 2\log d}\right\}}}{\left(\sqrt{\min\{k, 2\log d\}} + \sqrt{\frac{8\pi d}{d-1}} + \sqrt{\frac{8d}{d-1}\log\frac{dL}{\delta}}\right) \cdot \min\left\{5 + \sqrt{2\log\frac{ndL}{\delta}}, \sqrt{d}\right\}}, \quad (15)$$

then (7) holds, and hence NSN finds all correct neighbors for $y_1$ with probability at least $1 - \frac{3\delta}{1-\delta}$. The RHS of (15) is minimized when $k \geq 2\log d$, and consequently the condition (15) is equivalent to

$$\max\text{aff} < \frac{\sqrt{2\log d + \min\left\{2\log\left(\frac{n-d}{Cd}\right) - 4\log\log\frac{ne}{\delta}, \alpha\sqrt{d - 2\log d}\right\}}}{\left(\sqrt{2\log d} + \sqrt{\frac{8\pi d}{d-1}} + \sqrt{\frac{8d}{d-1}\log\frac{dL}{\delta}}\right) \cdot \min\left\{5 + \sqrt{2\log\frac{ndL}{\delta}}, \sqrt{d}\right\}}. \quad (16)$$

As $n$ is polynomial in $d$, there is a constant $C_3 > 0$ such that

$$(\text{RHS of (16)}) > \frac{C_3 \sqrt{\log{(n-d)} - \log\log{\frac{ne}{\delta}}}}{\sqrt{\log{\frac{dL}{\delta}} \cdot \log{\frac{ndL}{\delta}}}}$$

This completes the proof.

### B.4   Proof of Theorem 1

As we did in Section B.3, we prove in this section that if (1) holds then NSN finds all correct neighbors for $y_1$ with probability at least $1 - \frac{3\delta}{1-\delta}$.

The only difference between the semi-random model and the fully random model is the statistical dependence between subspaces. We can follow Step 3 in Section B.3 because they do not take any statistical dependence between subspaces into account. We assert that (7) is the success condition also for the fully random model. However, as $K = k_{max} = d$, there is no case where $k > k_{max}$ in this proof.

Now we provide a new proof of the last three steps for the fully random model.

#### Step 4: Lower bounds on the projection of correct points (the LHS of (7))

We again use Lemma 7. For $k > d/2$, we use the fact that $\|V_k^\top x_{j_k^*}\|_2$ stochastically dominates $P_{\lfloor d/2 \rfloor, (k)}$. Then it follows from Lemma 5a that

$$\|V_k^\top x_{j_k^*}\|_2 \geq \frac{k}{2d} + \frac{1}{d}\min\left\{2\log\left(\frac{n-k+1}{Ck\left(\log{\frac{ne}{k\delta}}\right)^2}\right), \alpha\sqrt{d/2}\right\} \tag{17}$$

for all $k = 1, \ldots, d-1$ simultaneously with probability at least $1 - \frac{\delta}{1-\delta}$.

#### Step 5: Upper bounds on the projection of incorrect points (the RHS of (7))

We again use the notion of $X_k \in \mathbb{R}^{d \times k}$ which is defined in the proof of Theorem 2. Since $\|\text{Proj}_{\mathcal{V}_k} y_j\|_2 = \|V_k^\top D_1^\top y_j\|_2$, the RHS of (7) can be written as

$$\max_{j:j\in[N], w_j \neq 1} \|V_k^\top D_1^\top y_j\|_2 \tag{18}$$

Since the true subspaces are independent of each other, $y_j$ with $w_j \neq 1$ is also independent of $D_1$ and $V_k$, and its marginal distribution is uniform over $\mathbb{S}^{p-1}$. It follows from Lemma 5b that with probability at least $1 - \delta/d$,

$$(18) \leq \frac{\|V_k^\top D_1^\top\|_F}{\sqrt{p}} + \frac{\|V_k^\top D_1^\top\|_2}{\sqrt{p}} \cdot \sqrt{2\log\frac{n(L-1)e}{\delta/d}}$$

$$\leq \sqrt{\frac{k}{p}} + \sqrt{\frac{2}{p}\log\frac{ndLe}{\delta}}. \tag{19}$$

The last inequality is obtained using the facts $\|D_1 V_k\|_F = \sqrt{k}$ and $\|D_1 V_k\|_2 \leq 1$. The union bound provides that (19) holds for every $k = 1, \ldots, d-1$ with probability at least $1 - \delta$.

#### Final Step: Proof of the main theorem

Now it suffices to show that (17) > (19) for every $k = 1, 2, \ldots, d-1$, i.e.,

$$\sqrt{\frac{k}{2d} + \frac{1}{d}\min\left\{2\log\left(\frac{n-k+1}{Ck\left(\log{\frac{ne}{k\delta}}\right)^2}\right), \alpha\sqrt{\frac{d}{2}}\right\}} > \sqrt{\frac{k}{p}} + \sqrt{\frac{2}{p}\log\frac{ndLe}{\delta}}, \quad \forall k = 1, 2, \ldots, d-1. \tag{20}$$

where $\alpha, C$ are the constants described in Lemma 5a. (20) is equivalent to

$$\frac{d}{p} < \min_{1 \leq k \leq d-1} \frac{k/2 + \min\left\{ 2\log\left(\frac{n-k+1}{Ck}\right) - 4\log\left(\log\frac{ne}{k\delta}\right), \alpha\sqrt{d/2} \right\}}{\left(\sqrt{k} + \sqrt{2\log(ndL\delta^{-1}e)}\right)^2}. \tag{21}$$

As $n$ is polynomial in $d$, the numerator can be replaced by $O(k + \log(n - k + 1))$. The RHS is minimized when $k = O(\log(ndL\delta^{-1}))$. Hence, the above condition is satisfied if (1) holds.

## B.5 Proof of Lemma 7

We construct a generative model for two random variables that are equal in distribution to $\|V_k^\top x_{j_k^*}\|_2^2$ and $P_{k,(k)}^2$. Then we show that the one corresponding to $\|V_k^\top x_{j_k^*}\|_2^2$ is greater than the other corresponding to $P_{k,(k)}^2$. This generative model uses the fact that for any isotropic distributions the marginal distributions of the components along any orthogonal axes are invariant.

The generative model is given as follows.

1. For $k = 1, \ldots, k_{max}$, repeat 2.

2. Draw $n - 1$ iid random variables $Y_1^{(k)}, \ldots, Y_{n-1}^{(k)}$ as follows.

$$Y_j^{(k)} = \left(1 - \sum_{i=1}^{k-1} Y_j^{(i)}\right) \cdot (X_{j1}^{(k)})^2, \ X_j^{(k)} \sim \mathrm{Unif}(\mathbb{S}^{d-k}), \quad \forall j = 1, \ldots, n-1.$$

   where $X_{j1}^{(k)}$ is the first coordinate of $X_j^{(k)}$. Define

$$\pi_k \triangleq \arg\max_{j:j \neq \pi_1, \ldots, \pi_{k-1}} \left(\sum_{i=1}^{k} Y_j^{(i)}\right).$$

3. For $k = k_{max} + 1, \ldots, d - 1$, repeat

$$\pi_k \triangleq \arg\max_{j:j \neq \pi_1, \ldots, \pi_{k-1}} \left(\sum_{i=1}^{k_{max}} Y_j^{(i)}\right).$$

We first claim that $(\sum_{i=1}^{k} Y_{\pi_k}^{(i)})$ is equal in distribution to $\|V_k^\top x_{j_k^*}\|_2^2$. Consider the following two sets of random variables.

$$A_k \triangleq \left(\sum_{i=1}^{k} Y_j^{(i)} : j \in [n-1], j \neq \pi_1, \ldots, \pi_{k-1}\right),$$

$$B_k \triangleq \left(\|V_k^\top x_j\|_2^2 : w_j = 1, j \neq 1, j_1^*, \ldots, j_{k-1}^*\right).$$

Each set contains $(n - k)$ random variables. We prove by induction that the joint distribution of the random variables of $A_k$ is equal to those of $B_k$. Then the claim follows because $(\sum_{i=1}^{k} Y_{\pi_k}^{(i)})$ and $\|V_k^\top x_{j_k^*}\|_2^2$ are the maximums of $A_k$ and $B_k$, respectively.

- Base case : As $V_1 = x_1$, $B_1 = (\|V_1^\top x_j\|_2^2 : w_j = 1, j \neq 1)$ is the set of squared inner products with $x_1$ for the $n-1$ other points. Since the $n-1$ points are iid uniformly random unit vectors independent of $x_1$, the squared inner products with $x_1$ are equal in distribution to $Y_j^{(1)} = (X_{j1}^{(1)})^2$. Therefore, the joint distribution of $B_1 = (\|V_1^\top x_j\|_2^2 : w_j = 1, j \neq 1)$ is equal to the joint distribution of $A_1 = (Y_j^{(1)} : j = 1, \ldots, n-1)$.

- Induction : Assume that the joint distribution of $A_k$ is equal to the joint distribution of $B_k$. It is sufficient to show that given $A_k \equiv B_k$ the conditional joint distribution of $A_{k+1} = (\sum_{i=1}^{k+1} Y_j^{(i)} : j \in [n-1], j \neq \pi_1, \ldots, \pi_k)$ is equal to the conditional joint distribution of $B_{k+1} = (\|V_{k+1}^\top x_j\|_2^2 : w_j = 1, j \neq 1, j_1^*, \ldots, j_k^*)$. Define

$$v_k = \frac{x_{j_k^*} - V_k V_k^\top x_{j_k^*}}{\|x_{j_k^*} - V_k V_k^\top x_{j_k^*}\|_2}.$$

$v_k$ is the unit vector along the new orthogonal axis added on $\mathcal{V}_k$ for $\mathcal{V}_{k+1}$. Since we have

$$\|V_{k+1}^\top x_j\|_2^2 = \|V_k^\top x_j\|_2^2 + (v_k^\top x_j)^2, \quad \forall j : w_j = 1,$$

The two terms are independent of each other because $V_k \perp v_k$, and $x_j$ is isotropically distributed. Hence, we only need to show that $((v_k^\top x_j)^2 : w_j = 1, j \neq 1, j_1^*, \dots, j_k^*)$ is equal in distribution to $(Y_j^{(k+1)} : j \in [n-1], j \neq \pi_1, \dots, \pi_k)$.

Since $v_k$ is a normalized vector on the subspace $\mathcal{V}_k^\perp \cap \mathcal{D}_1$, and $x_{j_k^*}$ is drawn iid from an isotropic distribution, $v_k$ is independent of $V_k^\top x_{j_k^*}$. Hence, the marginal distribution of $v_k$ given $\mathcal{V}_k$ is uniform over $(\mathcal{V}_k^\perp \cap \mathcal{D}_1) \cap \mathbb{S}^{p-1}$. Also, $v_k$ is also independent of the points $\{x_j : w_j = 1, j \neq 1, j_1^*, \dots, j_k^*\}$. Therefore, the random variables $(v_k^\top x_j)^2$ for $j$ with $w_j = 1, j \neq 1, j_1^*, \dots, j_k^*$ are iid equal in distribution to $Y_j^{(k+1)}$ for any $j$.

Second, we can see that the $k$'th maximum of $\{\sum_{i=1}^k Y_j^{(i)} : j \in [n-1]\}$ is equal in distribution to $P_{k,(k)}^2$. This is because each $\sum_{i=1}^k Y_j^{(i)}$ can be seen as the norm of the projection of a uniformly random unit vector in $\mathbb{R}^d$ onto a $k$-dimensional subspace.

Now we are ready to complete the proof. Since $\left(\sum_{i=1}^k Y_{\pi_k}^{(i)}\right)$ is the maximum of the $n-k$ variables of $A_k$, it is greater than or equal to the $k$'th maximum of $\left(\sum_{i=1}^k Y_j^{(i)} : j \in [n-1]\right)$. Therefore, $\|V_k^\top x_{j_k^*}\|_2^2$ stochastically dominates $P_{k,(k)}^2$.

The second claim is clear because $\mathcal{V}_{\hat{k}} \subseteq \mathcal{V}_k$, and hence the norm of the projection onto $\mathcal{V}_k$ is always larger than the norm of the projection onto $\mathcal{V}_{\hat{k}}$.

### B.6   Proof of Lemma 5a

Let $x$ be an unit vector drawn uniformly at random from $\mathbb{S}^{d-1}$. Equivalently, $x$ can be drawn from

$$x = \frac{w}{\|w\|_2}, \quad w \sim \mathcal{N}(0, I_{d \times d}).$$

Define $A^\perp \in \mathbb{R}^{(d-k) \times d}$ as a matrix with orthonormal rows such that $\|w\|_2^2 = \|Aw\|_2^2 + \|A^\perp w\|_2^2$ for any $w \in \mathbb{R}^d$. We have

$$
\begin{aligned}
\Pr\left\{\|Ax\|_2^2 > \frac{k}{d}(1+\epsilon)\right\} &= \Pr\left\{\frac{\|Aw\|_2^2}{\|w\|_2^2} > \frac{k}{d}(1+\epsilon)\right\} \\
&= \Pr\left\{\frac{\|Aw\|_2^2}{\|Aw\|_2^2 + \|A^\perp w\|_2^2} > \frac{k}{d}(1+\epsilon)\right\} \\
&\geq \Pr\left\{\|Aw\|_2^2 > k(1+\epsilon), \|A^\perp w\|_2^2 < (d-k) - k\epsilon\right\} \\
&= \Pr\left\{\|Aw\|_2^2 > k(1+\epsilon)\right\} \cdot \Pr\left\{\|A^\perp w\|_2^2 < (d-k) - k\epsilon\right\}, \quad (22)
\end{aligned}
$$

where the last equality follows from that $\|Aw\|_2$ and $\|A^\perp w\|_2$ are independent of each other because $w \sim \mathcal{N}(0, I_{d \times d})$. Note that $\|Aw\|_2^2$ and $\|A^\perp w\|_2^2$ are Chi-square random variables with degrees of freedom $k$ and $d-k$, respectively.

Now we use the following lemma.

**Lemma 8 (Chi-square upper-tail lower-bound)**  *For any $k \in \mathbb{N}$ and any $\epsilon \geq 0$, we have*

$$\Pr\{\chi_k^2 \geq k(1+\epsilon)\} \geq \frac{1}{3\sqrt{k\epsilon} + 6} \exp\left(-\frac{k\epsilon}{2}\right).$$

*where $\chi_k^2$ is the chi-square random variable with $k$ degrees of freedom.*

Suppose $0 \leq \epsilon \leq \alpha \frac{(d-k)^{\frac{1}{2}}}{k}$ for some $\alpha \in (0,1)$. It follows from Lemma 8 and the central limit theorem that

$$(22) \geq \Pr\left\{\|Aw\|_2^2 > k(1+\epsilon)\right\} \cdot \Pr\left\{\|A^\perp w\|_2^2 - (d-k) < -\alpha(d-k)^{\frac{1}{2}}\right\}$$

$$\geq \frac{f(\alpha)}{3k\epsilon + 6} \exp\left(-\frac{k\epsilon}{2}\right)$$

where $f(\alpha) \in (0,1)$ is some constant depending only on $\alpha$.

Then it follows that

$$\Pr\left\{z_{(n-m+1)}^2 < \frac{k}{d}(1+\epsilon)\right\} = \Pr\left\{\exists I \subset [n], |I| = n - m + 1 : z_i^2 < \frac{k}{d}(1+\epsilon), \forall i \in I\right\}$$

$$\leq \binom{n}{m-1} \cdot \Pr\left\{z_1^2 < \frac{k}{d}(1+\epsilon)\right\}^{n-m+1}$$

$$\leq \left(\frac{ne}{m}\right)^m \cdot \left(1 - \frac{f(\alpha)}{3k\epsilon + 6} \exp\left(-\frac{k\epsilon}{2}\right)\right)^{n-m+1}$$

$$\leq \exp\left\{m \log \frac{ne}{m} - \frac{f(\alpha) \cdot (n-m+1)}{3k\epsilon + 6} \exp\left(-\frac{k\epsilon}{2}\right)\right\} \quad (23)$$

where we use the facts $\binom{n}{m} \leq \left(\frac{ne}{m}\right)^m$ and $1 + x \leq e^x, \forall x$.

Set $C = \frac{6}{f(\alpha)}$, and choose $\epsilon$ such that

$$\epsilon = \frac{1}{k} \min\left\{2 \log\left(\frac{n-m+1}{Cm\left(\log \frac{ne}{m\delta}\right)^2}\right), \alpha\sqrt{d-k}\right\}.$$

This $\epsilon$ is valid because $0 \leq \epsilon \leq \alpha \frac{(d-k)^{\frac{1}{2}}}{k}$. Then we obtain

$$(23) \leq \exp\left\{m \log \frac{ne}{m} - \frac{f(\alpha) \cdot (n-m+1)}{6 \log\left(\frac{n-m+1}{Cm\left(\log \frac{ne}{m\delta}\right)^2}\right) + 6} \cdot \frac{Cm\left(\log \frac{ne}{m\delta}\right)^2}{n-m+1}\right\}$$

$$= \exp\left\{m \log \frac{ne}{m} - \frac{6 \log \frac{ne}{m\delta}}{6\left(1 + \log\left(\frac{f(\alpha)}{6} \cdot \frac{n-m+1}{m} \cdot (\log \frac{ne}{m\delta})^{-2}\right)\right)} \cdot m \log \frac{ne}{m\delta}\right\}$$

$$\leq \exp\left\{m \log \frac{ne}{m} - \frac{6(1 + \log \frac{n}{m})}{6(1 + \log \frac{f(\alpha)}{6} + \log \frac{n}{m})} m \log \frac{ne}{m\delta}\right\}$$

$$\leq \exp\left\{m \log \frac{ne}{m} - m \log \frac{ne}{m\delta}\right\}$$

$$\leq \delta^m.$$

This completes the proof.

## B.7  Proof of Lemma 5b

We use a special case of Levy's lemma for this proof.

**Lemma 9 ([12])** *For $x \sim \text{Unif}(\mathbb{S}^{d-1})$,*

$$\Pr\{\|Ax\|_2 > m\|Ax\|_2 + t\} \leq \exp\left(-\frac{dt^2}{2\|A\|_2^2}\right),$$

$$\Pr\{\|Ax\|_2 < m\|Ax\|_2 - t\} \leq \exp\left(-\frac{dt^2}{2\|A\|_2^2}\right).$$

*for any matrix $A \in \mathbb{R}^{p \times d}$ and $t > 0$. $m\|Ax\|_2$ is the median of $\|Ax\|_2$.*

It follows from the lemma that

$$|E\|Ax\|_2 - m\|Ax\|_2| \le E\left[|\|Ax\|_2 - m\|Ax\|_2|\right] \le \int_0^\infty 2e^{-\frac{dt^2}{2\|A\|_2^2}}\,dt = \sqrt{\frac{2\pi}{d}}\|A\|_2.$$

Then we have

$$\Pr\left\{\|Ax_i\|_2 > \sqrt{\frac{\|A\|_F^2}{d}} + \sqrt{\frac{2\pi}{d}}\|A\|_2 + t\right\} = \Pr\left\{\|Ax_i\|_2 > \sqrt{E\|Ax_i\|_2^2} + \sqrt{\frac{2\pi}{d}}\|A\|_2 + t\right\}$$

$$\le \Pr\left\{\|Ax_i\|_2 > E\|Ax_i\|_2 + \sqrt{\frac{2\pi}{d}}\|A\|_2 + t\right\}$$

$$\le \Pr\left\{\|Ax_i\|_2 > m\|Ax_i\|_2 + t\right\}$$

$$\le \exp\left(-\frac{dt^2}{2\|A\|_2^2}\right).$$

It follows that

$$\Pr\left\{z_{(n-m+1)} > \sqrt{\frac{\|A\|_F^2}{d}} + \sqrt{\frac{2\pi}{d}}\|A\|_2 + t\right\}$$

$$\le \Pr\left\{\exists I \subset [n], |I| = m : \|Ax_i\|_2 > \sqrt{\frac{\|A\|_F^2}{d}} + \sqrt{\frac{2\pi}{d}}\|A\|_2 + t, \forall i \in I\right\}$$

$$\le \binom{n}{m} \cdot \Pr\left\{\|Ax_1\|_2 > \sqrt{\frac{\|A\|_F^2}{d}} + \sqrt{\frac{2\pi}{d}}\|A\|_2 + t\right\}^m$$

$$\le \left(\frac{ne}{m}\right)^m \cdot \exp\left(-\frac{mdt^2}{2\|A\|_2^2}\right)$$

$$= \exp\left\{m\log\frac{ne}{m} - \frac{mdt^2}{2\|A\|_2^2}\right\}.$$

Replacing $t$ with $\sqrt{\frac{2\|A\|_2^2}{d}\log\frac{ne}{m\delta}}$, we obtain the desired result.

## B.8 Proof of Lemma 6a

Let $A = U\Sigma V^\top$ be the singular value decomposition of $A$. Then we have

$$E[\|AX\|_F^2] = E[\|U\Sigma V^\top X\|_F^2] = E[\|\Sigma X\|_F^2] = \sum_{i=1}^{\min(p,d)}\sigma_i^2\cdot\left(\sum_{j=1}^{k}E[X_{ij}^2]\right) = \sum_{i=1}^{\min(p,d)}\sigma_i^2\cdot\frac{k}{d} = \frac{k}{d}\|A\|_F^2.$$

where the second last equality follows from that $X_{ij}$ is a coordinate of a uniformly random unit vector, and thus

$$E[X_{ij}^2] = \frac{1}{d}, \quad \forall i, j.$$

## B.9 Proof of Lemma 6b

Consider the Stiefel manifold $V_k(\mathbb{R}^d)$ equipped with the Euclidean metric. We see that $X$ is drawn from $V_k(\mathbb{R}^d)$ with the normalized Harr probability measure. We have

$$\|AX\|_F - \|AY\|_F \le \|AX - AY\|_F = \|A(X-Y)\|_F \le \|A\|_2\|X-Y\|_F$$

for any $X, Y \in \mathbb{R}^{d\times k}$. Since $\|A\|_2 \le 1$, $\|AX\|_F$ is a 1-Lipschitz function of $X$. Then it follows from [15, 12] that

$$\Pr\{\|AX\|_F > m\|AX\|_F + t\} \le e^{-\frac{(d-1)t^2}{8}},$$

where $m\|AX\|_F$ is the median of $\|AX\|_F$. Also, we have

$$\Pr\{|\|AX\|_F - m\|AX\|_F| > t\} \leq 2e^{-\frac{(d-1)t^2}{8}},$$

and then it follows that

$$|E\|AX\|_F - m\|AX\|_F| \leq E\left[|\|AX\|_F - m\|AX\|_F|\right] \leq \int_0^\infty 2e^{-\frac{(d-1)t^2}{8}} dt = \sqrt{\frac{8\pi}{d-1}}.$$

It follows from Jensen's inequality and Lemma 6a that

$$E\|AX\|_F \leq \sqrt{E\|AX\|_F^2} = \sqrt{\frac{k}{d}}\|A\|_F$$

Putting the above inequalities together using the triangle inequality, we obtain the desired result.

## B.10 Proof of Lemma 8

For $k \geq 2$, it follows from [9, Proposition 3.1] that

$$
\begin{aligned}
\Pr\{\chi_k^2 \geq k(1+\epsilon)\} &\geq \frac{1-e^{-2}}{2} \frac{k(1+\epsilon)}{k\epsilon + 2\sqrt{k}} \exp\left(-\frac{1}{2}(k\epsilon - (k-2)\log(1+\epsilon) + \log k)\right) \\
&\geq \frac{1}{3\sqrt{k}\epsilon + 6} \exp\left(-\frac{k}{2}(\epsilon - \log(1+\epsilon))\right) \\
&\geq \frac{1}{3\sqrt{k}\epsilon + 6} \exp\left(-\frac{k\epsilon}{2}\right).
\end{aligned}
$$

For $k = 1$, we can see numerically that the inequality holds.