[Reviews · NeurIPS 2014]

Submitted by Assigned_Reviewer_2

The paper considers the problem of subspace clustering where the goal is to jointly find the spaces and the subset of points belonging to each. As pointed out in the related work, this area has seen very interesting progress recently starting with the SSC algorithm Elhamifar and Vidal using l1 minimization.

Most of these methods are done in two steps, first one finds a neighborhood matrix indicating which points belong in the same subspace, and then performing some sort of clustering to find the subspaces.

Here the authors propose a greedy approach for both steps. To find the neighborhood of a point i, the point j closest to i is added first, followed by the point closest to the span of i and j etc. Once the neighborhood matrix is constructed, one can either perform spectral clustering or use another greedy procedure the authors describe to find the subspaces.

Interestingly this procedure has very nice theoretical properties (analogous in some sense to how orthogonal matching pursuit often performs just as well as lasso under certain conditions). Compared to most methods which just prove the correctness of the first step (picking neighborhoods) here the authors prove the correctness of both steps (i.e. exact clustering). They allow the subspaces to intersect (controlled by some max affinity).

Interestingly, they show that as the number of points grows a higher max affinity is allowed for their approach (as one would like). This is different from another greedy approach TSC.

The experiments compare the authors approach to SSC (l1 minimization) and two greedy approaches GFS and TSC. The new approach performs better than the other greedy approaches while being comparable to SSC. (Of course being greedy it is much faster than SSC).

Quality: Paper is thorough, authors provide a novel algorithm, theory and experiments.

Clarity: I found the work to be very well written and easy to understand.

Originality: Good, not incremental but also not revolutionary

Significance: Moderate, again not incremental, but also not revolutionary.

Summary: The paper presents a greedy algorithm for subspace clustering that has strong theoretical guarantees of correctness. Empirically the method performs better than other greedy approaches. The paper is quite interesting and very well written.

Submitted by Assigned_Reviewer_4

The paper was interesting to read and was mostly well presented given the space constraints. However, the work could have been better motivated at the start. The application and experimental section only explained briefly and the supplement could add further information on these.

The paper didn't really explain the importance of subspace clustering very well.

Abstract: Written in a vague way for the casual reader. Could be more precise.

Page 2. Line -3. In the text you say "union" but the equation has intersection.

Page 5. Theorem 1. The conditions on n are hard to disentangle because the n is on both sides of the inequality. Could a bound on n (possibly less tight) be given too to give a better impression of the constraint?

Table 2 shows the results for various values of L. How can the true number of clusters/subspaces be estimated in practice?

Further guidance on the choice of K needed too.
Summary: The paper was interesting to read and was mostly well presented given the space constraints. However, the work could have been better motivated at the start. The application and experimental section only explained briefly and the supplement could add further information on these.

Submitted by Assigned_Reviewer_30

This paper presents a new method for subspace clustering. The method includes a set of theoretical guarantees, as well as algorithms to implement the method.

The method is well-motivated and addresses an interesting clustering problem. While I am not hugely familiar with the literature on subspace clustering, it seems to me that there is significant substance to this paper.

My main criticisms are that the discussion section is very short (more detailed discussions would be very interesting/useful), and that a table of speed comparisons with the other methods would be very useful to see. I would personally also like to see some comparisons of clustering quality with standard (non-subspace) methods such as Kmeans. Of course, I'd expect the subspace clustering methods to outperform Kmeans, but it would be very interesting to see by how much this is the case.

It would also be nice to see a suggestion as to how to best infer the number of clusters. This is a common challenge for many clustering methods, and is very useful when using a given method in practice.

Minor points:

Adjusted Rand Index might also be a useful metric for assessing cluster quality.
Summary: An interesting paper on subspace clustering, with significant content. Needs a bit more work in a couple of areas (esp discussion of results) to really get the most out of what is very interesting underlying theoretical work.
Author Feedback
Author rebuttal: We appreciate the reviewers for their valuable and encouraging comments.

Response to reviewer_30:
---------
1. The discussion section
Though we had to shorten the discussion due to the page limit, we have had a few more points to discuss, such as the estimation of the cluster number, as mentioned by the reviewer, and the robustness issue, etc. We will discuss these in the final version.

2. A table of speed comparisons
We have compared the speeds of different subspace clustering methods and neighborhood construction methods for synthetic data. The result was similar to the elapsed times in Table 2. We will add some plots of speed comparisons in different parameter settings in the supplementary material.

3. Comparisons with standard clustering algorithms
We have not run standard clustering algorithms in the experiment. We agree that our method (as well as the other subspace clustering algorithms) will outperform Kmeans. We will also run Kmeans to compare and provide the results to demonstrate the expectation in the final version.

4. Inferring the number of clusters
In general, the number of clusters are estimated at the clustering step. Spectral clustering has its own way to estimate the number by finding a “knee” point, and GSR does not need to estimate the number of clusters a priori. Once the algorithms finishes, the number of the resulting groups will be our estimate of L.

5. Adjusted Rand Index (ARI)
ARI will be very good measure to compare our algorithm with the existing ones. We believe that our measure of Clustering Error (CE) can measure the clustering quality as well as ARI, but CE could not measure for a large numbers of clusters because of the optimization over permutations. We can try to use ARI for such cases.

Response to reviewer_4:
---------

1. Importance of subspace clustering
Due to the page limit, we referred the readers to a comprehensive survey [1]. In this survey paper, the author discusses many kinds of datasets in practice which can be modeled by unions of subspaces. For example, in Computer Vision, face images under different illumination conditions lie on a union of low-dimensional subspaces, each of which corresponds to one person. Subspace clustering algorithms can segment these images into groups with respect to the individuals. For another example, trajectories of feature points from a rigid-body motion lie on a low-dimensional subspace. When one is given a video clip of multiple objects in different motions, subspace clustering is the mostly used approach to segment the feature points with respect to the objects.

In the above applications, points in the same cluster are not always nearer than those in different clusters, in particular when the subspaces have intersection. While standard clustering algorithms will not perform properly, subspace clustering algorithms can segment the points with respect to the low-dimensional subspaces.

[1] R. Vidal, “Subspace clustering", IEEE Signal Processing Magazine, vol. 28, no. 2, March 2011.

2. Abstract
We will rewrite the abstract more clearly in the final version.

3. Page 2. Line -3.
It should read “unions” of subspaces. Thank you for pointing out the typo.

4. Page 5, Theorem 1.
We would like to point out that the condition is satisfied when n is linear in d. We can rewrite the condition as:
n/d > C1 (log (n/d) + log(e/delta))^2,
which is satisfied when (n/d) is sufficiently large. Therefore, we can instead remove the (n/d) in the logarithm and put another constant C1’, which will be higher than the original C1, for the condition. We will try to remove the n in the RHS to make it easier to understand.

5. Estimating the number of clusters
In the experiments for Table 2, we assumed that L is known for all methods. When L is unknown, it can be estimated at the clustering step. For Spectral clustering, a well-known approach to estimate L is to find a “knee” point in the singular values of the neighborhood matrix. It is the point where the difference between two consecutive singular values are the largest. For GSR, we do not need to estimate the number of clusters a priori. Once the algorithms finishes, the number of the resulting groups will be our estimate of L.

6. The choice of K
The choice of K depends on the dimension of the subspaces d. If data points are lying exactly on the model subspaces, K=d is enough for GSR to recover a subspace. In practical situations where the points are near the subspaces, it is better to set K to be larger than d.

For simplicity of the paper, we have only considered the case where d is known. If d is unknown, or the subspaces are of different dimensions, a proper strategy should be addressed. For example, NSN can flexibly determine when to stop collecting neighbors based on the current neighbors.

We will discuss briefly in the final version.